# Elucidating the effect of drug-induced mitochondrial dysfunction on insulin signaling and glucose handling in skeletal muscle cell line (C2C12) *in vitro*

**Auxiliare Kuretu**[1], **Mamosheledi Mothibe**[1], **Phikelelani Ngubane**[2], **Ntethelelo Sibiya**[1] *

**1** Pharmacology Division, Faculty of Pharmacy, Rhodes University, Makhanda, South Africa, **2** School of Medical Sciences and Laboratory Medicine, University of KwaZulu-Natal, Durban, South Africa

* n.sibiya@ru.ac.za

**Data Availability Statement:** All relevant data are within the manuscript and its Supporting Information files

**Funding:** This study was supported by National Research Foundation: TTK220318196 awarded to

## Abstract

Efavirenz, tenofovir, rifampicin, simvastatin, lamotrigine and clarithromycin are known potential mitochondrial toxicants. Mitochondrial toxicity has been reported to disrupt the chain of events in the insulin signalling pathway. Considering the upward trajectory of diabetes mellitus prevalence, studies which seek to uncover probable risk factors for developing diabetes should be encouraged. This study aimed to evaluate the intracellular mechanisms leading to the development of insulin resistance in the presence of various conventional pharmacological agents reported as potential mitochondrial toxicants in skeletal muscle cell line. Differentiated C2C12 preparations were exposed to multiple concentrations of efavirenz, tenofovir, rifampicin, simvastatin, lamotrigine, and clarithromycin, separately. Glucose handling was evaluated by observing the changes in insulin-stimulated glucose uptake and assessing the changes in GLUT4 translocation, GLUT4 expression and Akt expression. The changes in mitochondrial function were evaluated by assessing mitochondrial membrane integrity, cellular ATP production, generation of intracellular reactive oxygen species, expression of tafazzin and quantification of medium malonaldehyde. Insulin stimulated glucose uptake was perturbed in C2C12 pre-treated with potential mitotoxicants. Additionally, ATP synthesis, alterations in mitochondrial membrane potential, excessive accumulation of ROS and malonaldehyde were observed in the presence of potential mitotoxicants. Particularly, we observed suppression of proteins involved in the insulin signalling pathway and maintenance of mitochondrial function namely GLUT4, Akt and tafazzin. Mitochondrial toxicants can potentially induce insulin resistance emanating from mitochondrial dysfunction. These new findings will contribute to the understanding of underlying mechanisms involved in the development of insulin resistance linked to mitochondrial dysfunction.

NS. https://www.nrf.ac.za/ The funders did not play any role in the study design, data collection and analysis, decision to publish, or preparation of the manuscript.

**Competing interests:** The authors have declared that no competing interests exist

## Introduction

Diabetes mellitus is a chronic metabolic disorder characterised by glucose homeostasis impairment resulting in hyperglycaemia [1–23]. The prevalence of type 2 diabetes (T2DM) is continuously increasing worldwide. According to the International Diabetes Federation (IDF), more than 90% of individuals with diabetes have T2DM [2, 3]. T2DM is driven by genetic, socio-economic, demographic and environmental factors. The aetiology of T2DM involves pancreatic ß cell dysfunction and insulin resistance in insulin-sensitive tissues such as hepatocytes, adipocytes and skeletal muscle [4–6]. Insulin resistance is associated with the accumulation of free fatty acids, elevated hepatic glucose production and decreased glucose uptake in insulin-sensitive tissues [4, 7, 8]. Compelling evidence suggests that mitochondrial impairment can potentially contribute to the development of insulin resistance [9–11]. Mitochondria play a significant role in cellular life, stress responses and death. Various studies have linked mitochondrial impairment with the initiation and propagation of inflammatory responses [12, 13]. Mitochondria have attracted the attention of researchers for decades and their importance in cellular function and energy homeostasis has spawned much interest in metabolic diseases such as a variety of cancers, cardiovascular diseases, and neurodegenerative disorders such as Parkinson's and Alzheimer's and T2DM [9, 12, 14]. Importantly, it has been suggested that mitochondrial dysfunction is associated with insulin resistance in skeletal muscle and other tissues such as the liver and adipose. Insulin resistance is associated with depleted levels of mitochondrial oxidative enzymes, ATP levels and abnormal mitochondrial morphology [9–11]. However, the relationship between mitochondrial dysfunction and the development of insulin resistance is not yet fully elucidated. Insistent evidence suggests that improving mitochondrial function can serve as a positive therapeutic tool in enhancing insulin sensitivity [10, 15]. Various drug classes have been reported to potentially induce insulin resistance ascribed to mitochondrial dysfunction. Drug classes such as statins, anti-diabetics, anti-epileptics, NSAIDs, anti-depressants, and certain antibiotics have been identified to induce mitochondrial toxicity and potentially cause insulin resistance however this hypothesis is not yet elucidated [15].

This study aimed to characterize the effect of selected potential mitotoxic drugs on glucose homeostasis in skeletal muscle cell lines *in vitro*. Skeletal muscle cells were utilized as they are the primary site of insulin-stimulated glucose uptake and play an essential role in maintaining glucose homeostasis [16–18]. Tenofovir, efavirenz, rifampicin, lamotrigine, clarithromycin, and simvastatin were the selected drugs of choice. These drugs have been indicated as potential mitochondrial toxicants that exert adverse effects on various biochemical pathways including the insulin signalling pathway [15, 19]. We envisaged that assessing changes in glucose uptake, GLUT 4 expression, GLUT 4 translocation, relative expression of Akt and tafazzin, ATP synthesis, oxidative stress status and mitochondrial membrane polarisation state would elucidate a direct link between drug-induced mitochondrial toxicity and impaired insulin sensitivity.

## Materials and methods

### Chemicals

The following chemicals, drugs and assay kits were purchased from Sigma-Aldrich (USA). Dulbecco's modified eagle's medium (DMEM), fetal bovine serum (FBS), and penicillin-streptomycin (USA, ATP bioluminescence Kit, Fluorometric Intracellular ROS Kit, Bovine Serum albumin (BSA), Tris HCL, Triton X100, Rabbit polyclonal anti-phospho-Akt antibody, Anti-Akt2 antibody produced in rabbit, GLUT4 antibody, Citric acid, Simvastatin, Clarithromycin, Rifampicin, Tenofovir, Efavirenz, Lamotrigine, 3-(4,5-dimethyl-2- thiazolyl)-2,5-diphenyl-

2H-tetrazolium bromide (MTT), Insulin, sodium phosphate dibasic ($Na_2HPO_4$), Monopotassium phosphate ($KH_2PO_4$), Sodium chloride (NaCl), Potassium chloride (KCl), Sodium hydroxide (NaOH), Hydrochloric acid (HCl), Tween 20, Paraformaldehyde, Dimethyl sulfoxide (DMSO), 3,3′,5,5′-Tetramethylbenzidine (TMB), Horseradish peroxidase (HRP), Hydrogen peroxide ($H_2O_2$), Trypan blue, Butanol, Ethanol Phosphoric acid ($H_3PO_4$), 2-Thiobarbituric acid (TBA), Butylated hydroxytoluene (BHT), 0.05% Trypsin–EDTA (Thermo-Fisher Scientific).

## Methods

### Skeletal muscle cell culture and differentiation

Assays were performed using skeletal muscle (C2C12), which were cultured in tissue culture flasks (T25 and T75) in the presence of Dulbecco's modified eagle's medium (DMEM) supplemented with 1% penicillin-streptomycin and 10% foetal bovine serum (FBS) in a humidified incubator with 5% $CO_2$ at 37˚C. After the cells had reached approximately 80% confluence, they were trypsinized and passaged into new flasks until ready for seeding.

### Cell viability

To evaluate cell viability, the 3-(4,5-dimethyl-2- thiazolyl)-2,5-diphenyl-2H-tetrazolium bromide (MTT) assay was performed in the presence of selected drugs. C2C12 cells were seeded into 96 well-plate clear bottom tissue culture plates at a density of ($4.65 \times 10^4$) until they reached approximately 80% confluence. The cells were exposed to different concentrations (25–400μM) of the drugs. After 24 hours, MTT solution (100μM) was added to each well, and the plate was incubated for three hours in the dark. Thereafter, DMSO (100μM) was added, and the plate was incubated for ten minutes. The absorbance was measured at 570nm using a UV-VIS Spectrophotometer. The cell viability was calculated as follows:

$$\text{Percentage viability}(\%) = \left( \frac{\text{Absorbance of sample}}{\text{Absorbance of control}} \right) \times 100$$

**NB:** The cell viability study demonstrated that the ideal concentrations for further biochemical analysis were (25–100 μM).

### Estimation of cellular glucose uptake

C2C12 were seeded at a density of ($4.35 \times 10^4$) in 24 well plates and were allowed to adhere. Thereafter, the myoblasts were differentiated into myotubes by switching normal medium to DMEM supplemented with 0.2% FBS and 1% penicillin-streptomycin for three days. The medium was replaced daily until differentiated multinucleated myotubes were observed under a microscope. The myotubes preparation were then preincubated at 37˚C for 24 hrs with different drug concentrations (25–100μM). After 24 hrs the culture medium was aspirated from the well and washed with Phosphate-Buffered Saline (200μM) three times. DMEM (200μL) supplemented with insulin (0.05μ/ml) was transferred into each well. The plate was incubated for a further 24 hours after which the medium glucose concentration was measured using an Accu-Check Performa glucometer. The cell preparation exposed to drug treatments but not treated with insulin served as a negative control, and preparations only treated with insulin served as a positive control. The assay was performed in triplicate. The estimation of glucose

uptake was determined using the following formula:

$$\text{Glucose uptake}(\%) = \frac{\text{Medium Glucose(T0)} - \text{Medium glucose(T24)}}{\text{Medium glucose(T0)}} \times 100$$

## In-cell ELISA

C2C12 cells were seeded at a density of $(4.97 \times 10^4)$ in 96 well plates. After the cells had reached approximately 80% confluence, they were preincubated at 37°C for 24 hrs with different drug concentrations (25–100 μM) in DMEM. After 24 hours, the medium was aspirated, and paraformaldehyde (8%, 100 μL) was added to each well for fixation. The plate was incubated for 15 minutes at room temperature whilst shaking with a microplate shaker at 300rpm. The paraformaldehyde was aspirated, and each well was washed four times using PBS (200 μL) followed by the addition of 2X permeabilization buffer (Triton X-100) (200 μL) into each well. Thereafter, the plate was incubated for 30 minutes at room temperature whilst shaking at 300 rpm. After 30 mins, the permeabilization buffer was aspirated from the wells and a blocking buffer (1X BSA, 200 μL) was added and incubated at room temperature for 2 hours whilst shaking the plate. Thereafter, the blocking buffer was aspirated and the relevant primary antibody (GLUT 4, Akt, Tafazzin) (100 μL) was added, separately. The plate was incubated at 4°C overnight. After incubation, the primary antibody was aspirated, and the plate was washed three times using wash buffer (250 μL). Thereafter, 100 μL of a secondary antibody (anti-rabbit IgG) with specificity to the primary antibody was added to each well. The plate was incubated at room temperature for 2 hours whilst shaking. After incubation, the plate was washed four times using wash buffer (250 μL) after which the horseradish peroxidase substrate (HRP) (100 μL)) was transferred into each well and incubated for 30 minutes whilst shaking. To stop the reaction, HCL (100 μL, 0.1M) was added into each well and absorbance was read at 450 nm using a spectrophotometer. The assays were performed in triplicate. To determine GLUT 4 expression, the same in-cell ELISA procedure was applied omitting the cell permeabilization step. The relative expression or translocation was calculated as follows:

$$\text{Relative expression}(\%) = \left( \frac{\text{Absorbance of treated wells}}{\text{Absorbance of control wells}} \right) \times 100$$

## Thiobarbituric acid reactive substances assay (TBARS)

A TBARS assay was performed by modifying a method developed by Potter et al to quantify malonaldehyde concentration (MDA) [20]. A medium sample (400 μL) was collected from treated C2C12 cells was transferred into a test tube containing 200 μL of 2% phosphoric acid ($H_3PO_4$). The homogenate was separated into 2 test tubes with each test tube receiving an equal volume (300 μL). Then 800 μL of TBA/BHT (TBA 0.67% w/v and BHT 0.01% w/v) solution was added into one test tube (sample test) and HCL (400 μL of 3 Mm) was added into the second test tube (blank). Both solutions were heated at 95°C for 20 minutes and were allowed to cool to room temperature. Butanol (1.5 mL) was added to the cooled samples. The sample solution was vortexed for 1 minute to ensure rigorous mixing and then allowed to settle until 2 distinguished liquid phases were observed. The butanol phase (top layer) was transferred to Eppendorf tubes and centrifuged at 13249 x g in a centrifuge for 15 minutes. The samples were aliquoted into wells of a 96-well microtiter plate in triplicates and the absorbance was read at 532 nm (reference λ 600 nm) using a microplate reader. The absorbance from these

wavelengths was used to calculate the concentration of MDA using the following formula which applies the Beer-Lambert law:

$$\text{MDA concentration} = \left( \frac{\text{Final absorbance} - \text{Blank}}{\text{MDA extinction coefficient} \left(156 \text{mmol}^{-1}\right)} \right)$$

## ATP bioluminescence assay

An ATP bioluminescence assay was performed according to the manufacturer's protocol (ATP Bioluminescence Assay Kit CN: FLAA-1KT, Saint Louis, USA) with the following adjustments. C2C12 cells were seeded in a solid black 96 well plate at a density of ($4.23 \times 10^4$) and incubated for 24 hours to allow the cells to adhere. Thereafter, the cells were treated with the drugs (clarithromycin, efavirenz, tenofovir, lamotrigine, simvastatin, and rifampicin) at different concentrations (25, 50 and 100 μM) for 24 hours. The control wells consisted of untreated cells. The medium was removed after 24 hours and luciferase (50μL) and luciferin (50μL) were added into each well. The plate was incubated at room temperature for 15 minutes and protected from light. The luminescence signal was detected using a microplate reader.

## Intracellular ROS assay

To detect ROS, a cell permeable reduced MitoTracker Red CM-H2XROS dye was used, and it produces fluoresces upon oxidation. An intracellular ROS fluorescence assay was performed according to the manufacturer's protocol (Fluorometric Intracellular ROS Kit CN: MAK143-1KT, Saint Louis, USA) with the following adjustments: Briefly, C2C12 cells were seeded in a solid black 96 well plate at a density of ($4.23 \times 10^4$) and incubated for 24 hours to allow the cells to adhere to the culture plates. The cells were treated with the drugs (clarithromycin, efavirenz, tenofovir, lamotrigine, simvastatin, and rifampicin) in concentrations of 25, 50 and 100 μM for 24 hours. The control wells consisted of untreated cells. Then 50 μL of the master reaction mix (ROS detection reagent) was pipetted into each well. The plate was incubated in a humidified incubator for one hour. The absorbance was set at an excitation wavelength of 490 nm and an emission wavelength of 525 nm using a microplate reader.

## Mitochondrial membrane potential assay

Mitochondrial inner membrane electrochemical potential changes were detected according to the manufacturer's protocol (Mitochondrial staining kit, CN: CS0390, Saint Louis, USA) with minor modifications. C2C12 cells were seeded in a solid black 96 well plate at a density of ($4.23 \times 10^4$) and incubated for 24 hours to allow the cells to adhere to the culture plates. The cells were treated with the drugs (clarithromycin, efavirenz, tenofovir, lamotrigine, simvastatin and rifampicin) at different concentrations (25, 50 and 100 μM) for 24 hours. The control had untreated wells. The growth medium was aspirated from the wells and the cells were overlayed with 50 μL of staining solution (JC-1 dye). The plate was incubated for 20 minutes at 37˚C. Fluorescence was determined at an excitation wavelength of 490nm and an emission wavelength of 525nm.

## Statistical analysis

Results were represented as mean ±SD. All the statistical analyses were performed using Graph Pad Prism (version 8.0.2). Assays were performed in triplicate and the mean was used for statistical analysis. Statistical significance was determined using the ordinary one-way ANOVA

test followed by Dunnett's post hoc test to compare the treatments and control. Significance was defined a priori as a $*P < 0.05$, $**P < 0.01$, $***P < 0.001$ and $****P < 0.0001$.

## Results

### Cell viability

A cytotoxicity study of mitochondrial toxicants was performed in C2C12 at concentrations of 25, 50, 100, 200 and 400 μM as displayed in Fig 1. All six drugs showed a significant statistical difference from the control at different concentrations except simvastatin and tenofovir at 25 μM and clarithromycin at 25 and 50 μM. The cell viability decreased as the concentration of the drugs increased.

### Glucose utilization

The percentage glucose uptake was estimated after treating C2C12 with mitochondrial toxicants (efavirenz, tenofovir, rifampicin, clarithromycin, simvastatin, lamotrigine) at different concentrations (25, 50, 100 μM) for 24 hours. The glucose uptake estimation is displayed in Fig 2. The insulin- treated group had a higher glucose uptake compared to the non-treated control. All the six drugs showed a concentration-dependent decrease in glucose uptake in comparison to the control. All the six drugs and insulin pre-treated group showed a significant statistical difference from the control at different concentrations except simvastatin and rifampicin at 25 μM and clarithromycin at 25 and 50 μM.

### GLUT4 translocation

Fig 3 displays the relative percentage of the GLUT4 translocation to the plasma membrane. GLUT 4 translocation gradually decreased in the presence of tenofovir, rifampicin, clarithromycin and simvastatin in comparison to the control. An acute increase in GLUT4 translocation was observed in C2C12 pre-treated with efavirenz and lamotrigine. There was a significant statistical difference in all six drugs at different concentrations except efavirenz at 50 and 100 μM and clarithromycin at 25 μM.

### GLUT 4 expression

Fig 4 indicates a decreased expression of GLUT4 protein in the presence of tenofovir, rifampicin, clarithromycin and simvastatin. An acute increase in GLUT4 expression was observed in C2C12 treated with efavirenz and simvastatin between concentrations of 25–100 μM. All six drugs showed a significant statistical difference from the control at different concentrations except for rifampicin at 25 and 50 μM.

### Akt expression

Fig 5 displays the relative percentage expression of Akt protein in C2C12 treated with mitotoxicants. As depicted in Fig 5, a decrease in Akt expression was observed with except in cells treated with lamotrigine where Akt expression was elevated in a directly proportional manner. All six drugs showed a significant statistical difference from the control at different concentrations except for simvastatin at 25 μM.

### Mitochondrial tafazzin expression analysis

Fig 6: displays the relative percentage expression of mitochondrial tafazzin protein in C2C12 treated with potential mitotoxicants. A decrease in tafazzin expression is observed in C2C12

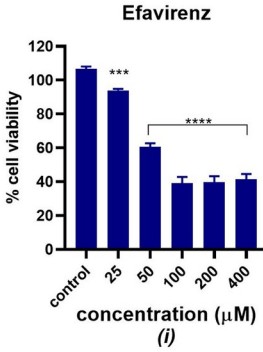

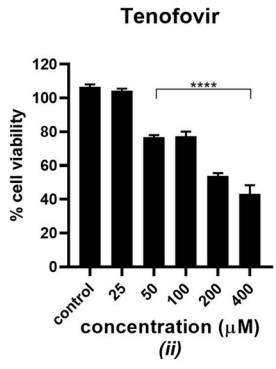

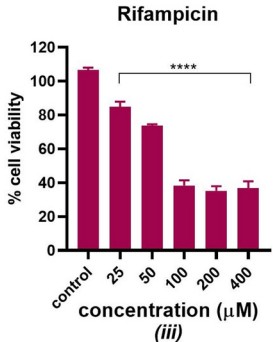

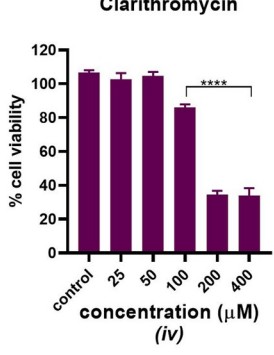

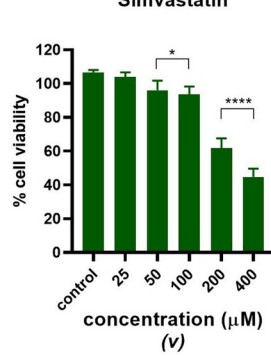

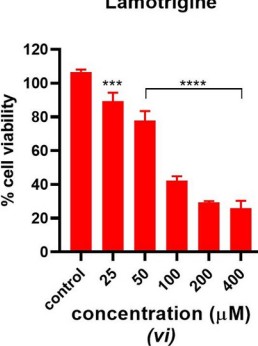

**Fig 1. Cell viability study after treating C2C12 with potential mitotoxicants for 24 hours at various concentrations (25–400μM).** The data are represented as mean ± SD represented with error bars, (n = 3), and the asterisk (*) represents the statistical difference between the test compounds and the control at (p < 0.05), (***) p<0.001, (****) p<0.0001.

pre-treated with simvastatin, lamotrigine, rifampicin, clarithromycin, efavirenz and tenofovir. All six drugs showed a significant statistical difference from the control at different concentrations except for (rifampicin, efavirenz) at 25 μM and (clarithromycin, lamotrigine) at 25 and 50 μM.

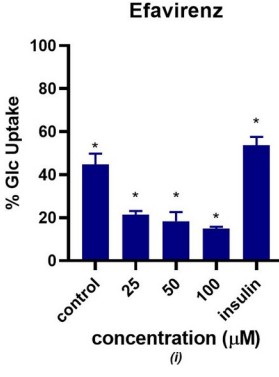

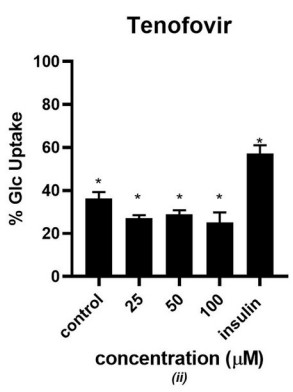

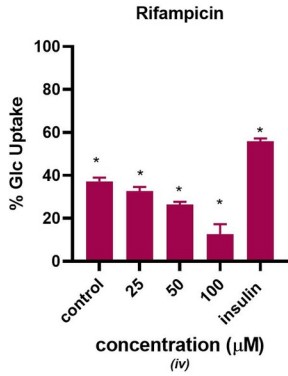

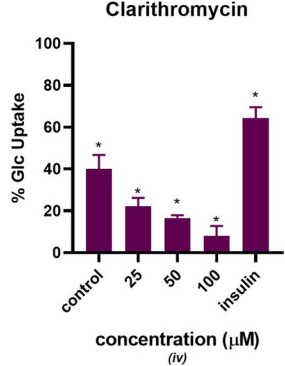

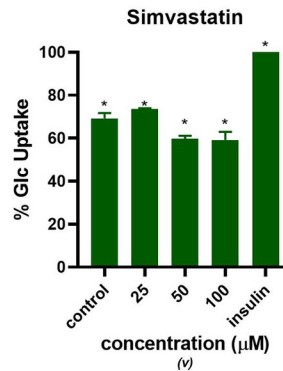

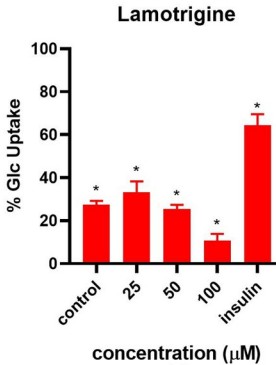

**Fig 2. Percentage glucose uptake estimation after treating C2C12 with potential mitotoxicants at 25, 50 and 100 μM for 24 hours.** The data are represented as mean ± SD represented with error bars, (n = 3), and the asterisk (*) represents the statistical difference between the test compounds and the control at $p < 0.05$, (**) $p < 0.01$, (***) $p < 0.001$, (****) $p < 0.001$.

## Intracellular reactive oxygen species

Fig 7 displays a significant increase in fluorescence intensity in the presence of simvastatin, lamotrigine, clarithromycin and efavirenz indicating increased oxidation of MitoTracker Red by ROS. However, rifampicin and tenofovir acutely increased the fluorescence intensity. A

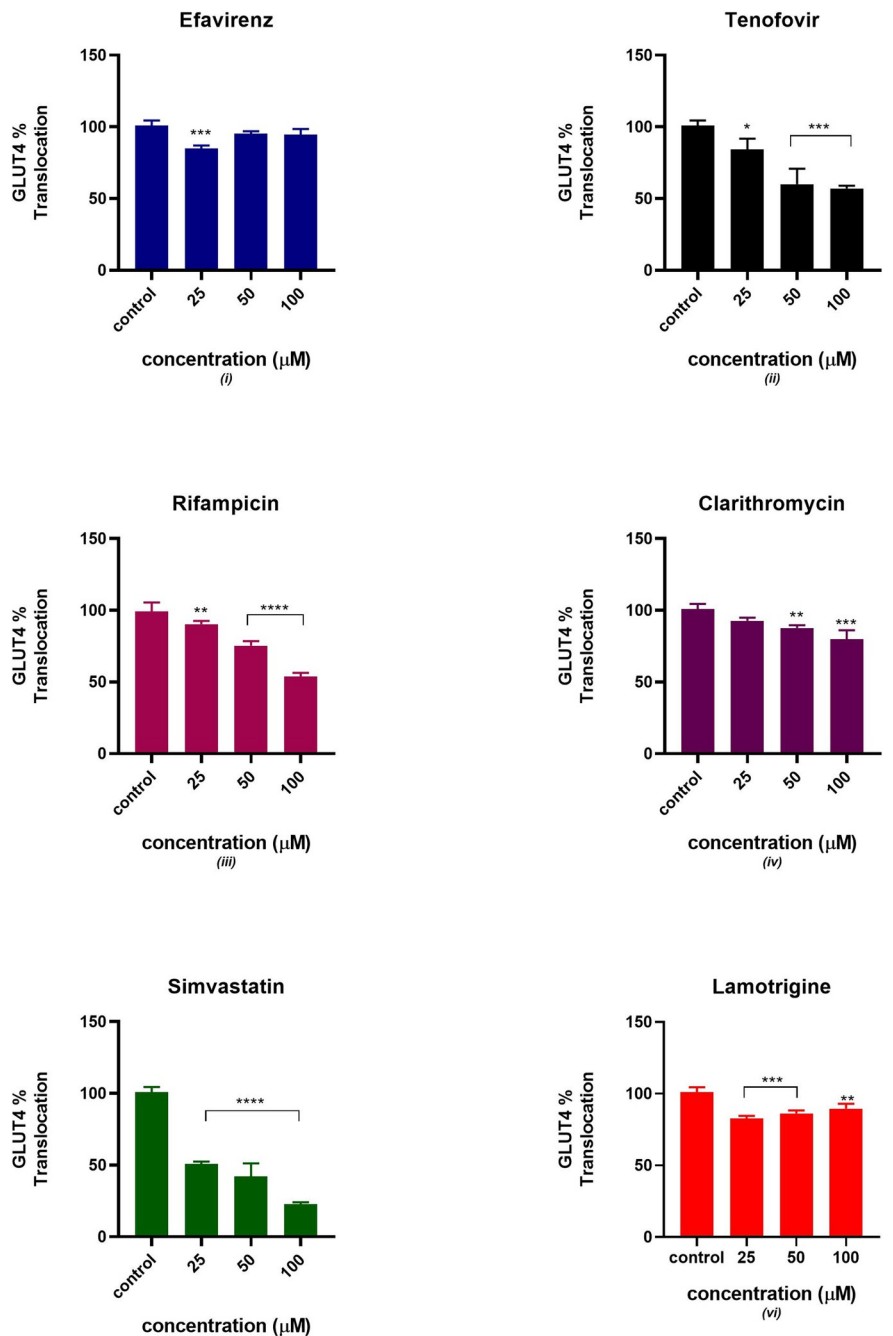

**Fig 3. Relative GLUT 4% translocation after 24 hours of exposure to potential mitotoxicants (25, 50, 100 μM).** The data are represented as mean ± SD represented with error bars, (n = 3), and the asterisk (*) represents the statistical difference between the test compounds and the control at (p < 0.05), (**) p<0.01, (***) p<0.001, (****) p<0.0001.

directly proportional relationship between drug concentration and fluorescence intensity was observed. There was no significant statistical difference between tenofovir and the control at p < 0.05. All the remaining drugs showed a significant statistical difference from the control at different concentrations except for clarithromycin at 50 μM and lamotrigine at 25 μM.

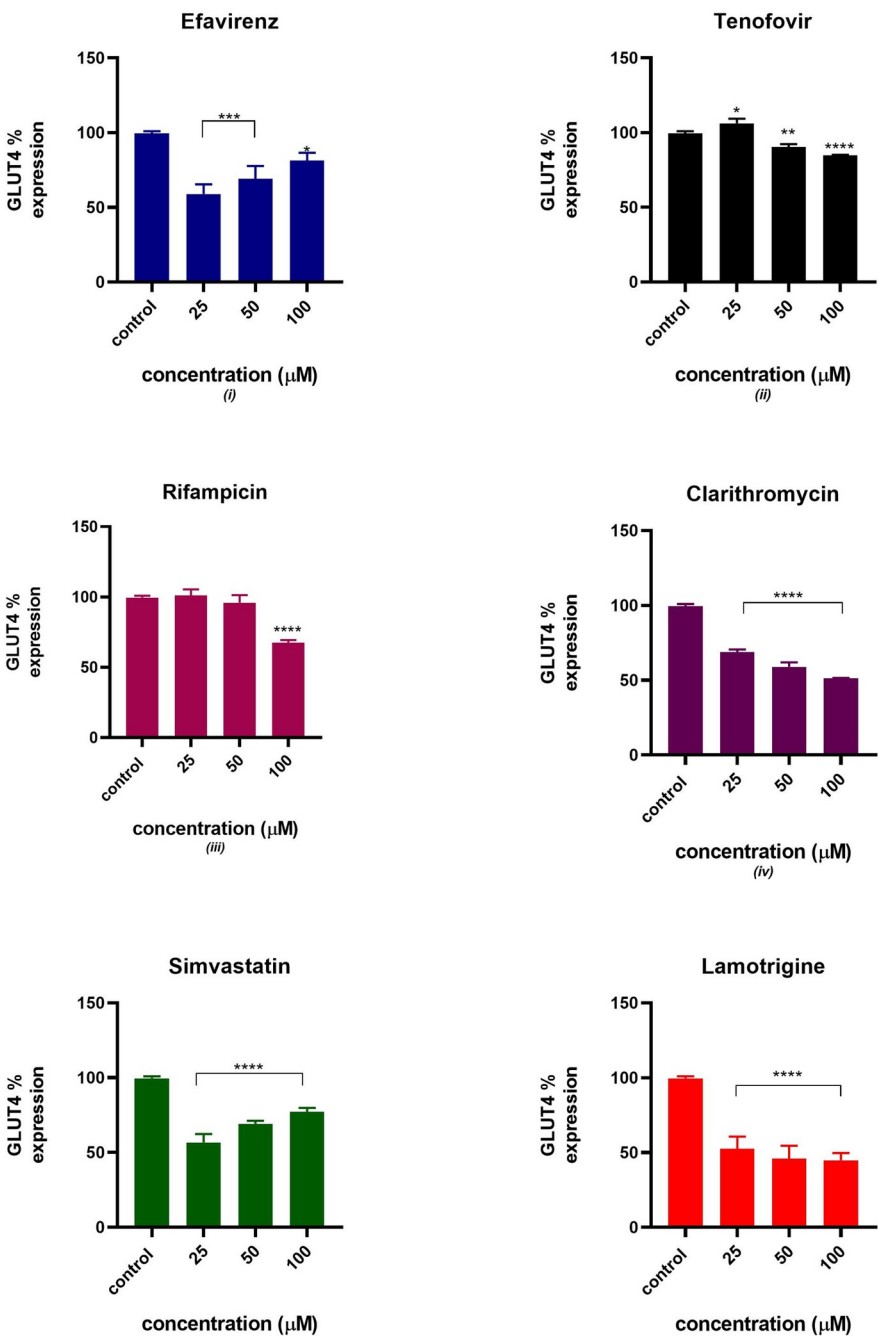

**Fig 4. Relative GLUT4% expression after 24 hours of exposure to potential mitotoxicants (25, 50, 100 μM).** The data are represented as mean ± SD represented with error bars, (n = 3), and the asterisk (*) represents the statistical difference between the test compounds and the control at (p < 0.05), (**) p<0.01, (***) p<0.001, (****) p<0.0001.

## Lipid peroxidation

Fig 8 indicates a dose-dependent increase in the medium MDA in the presence of five drugs except for tenofovir which gradually decreased the MDA concentration. There was a significant statistical difference from the control in all the drugs at different concentrations except for rifampicin at 25 μM.

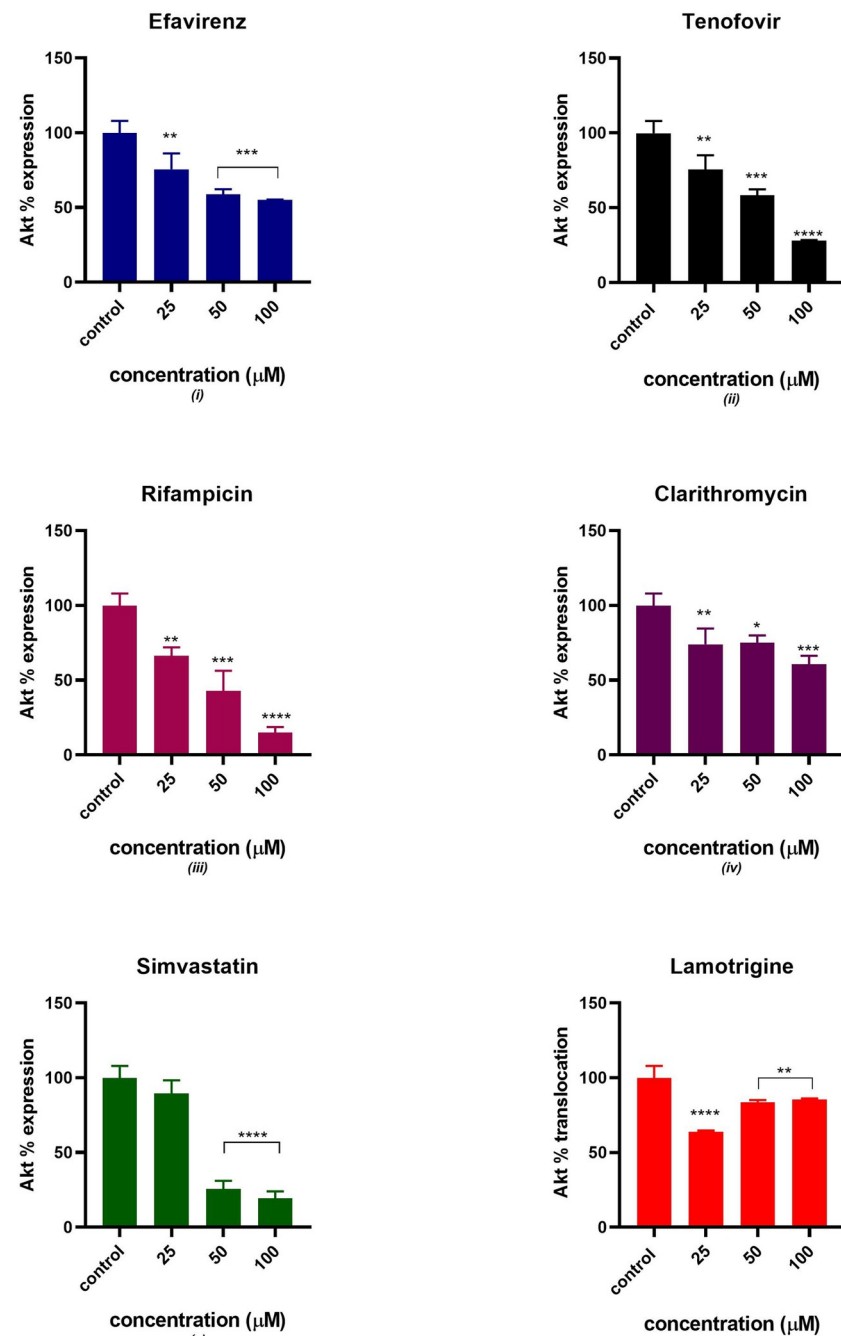

**Fig 5. Relative Akt % expression after 24 hours of exposure to potential mitotoxicants (25, 50, 100 μM).** The data are represented as mean ± SD represented with error bars, (n = 3), and the asterisk (*) represents the statistical difference between the test compounds and the control at ($p < 0.05$), (**) $p < 0.01$, (***) $p < 0.001$, (****) $p < 0.0001$.

## Mitochondrial membrane potential

Fig 9 depicts an acute reduction in mitochondrial membrane potential observed in the presence of rifampicin and simvastatin. However, efavirenz and lamotrigine slightly increased the membrane potential. There was a significant increase in mitochondrial membrane potential in the presence of tenofovir and clarithromycin. The control consisted of untreated cells. There

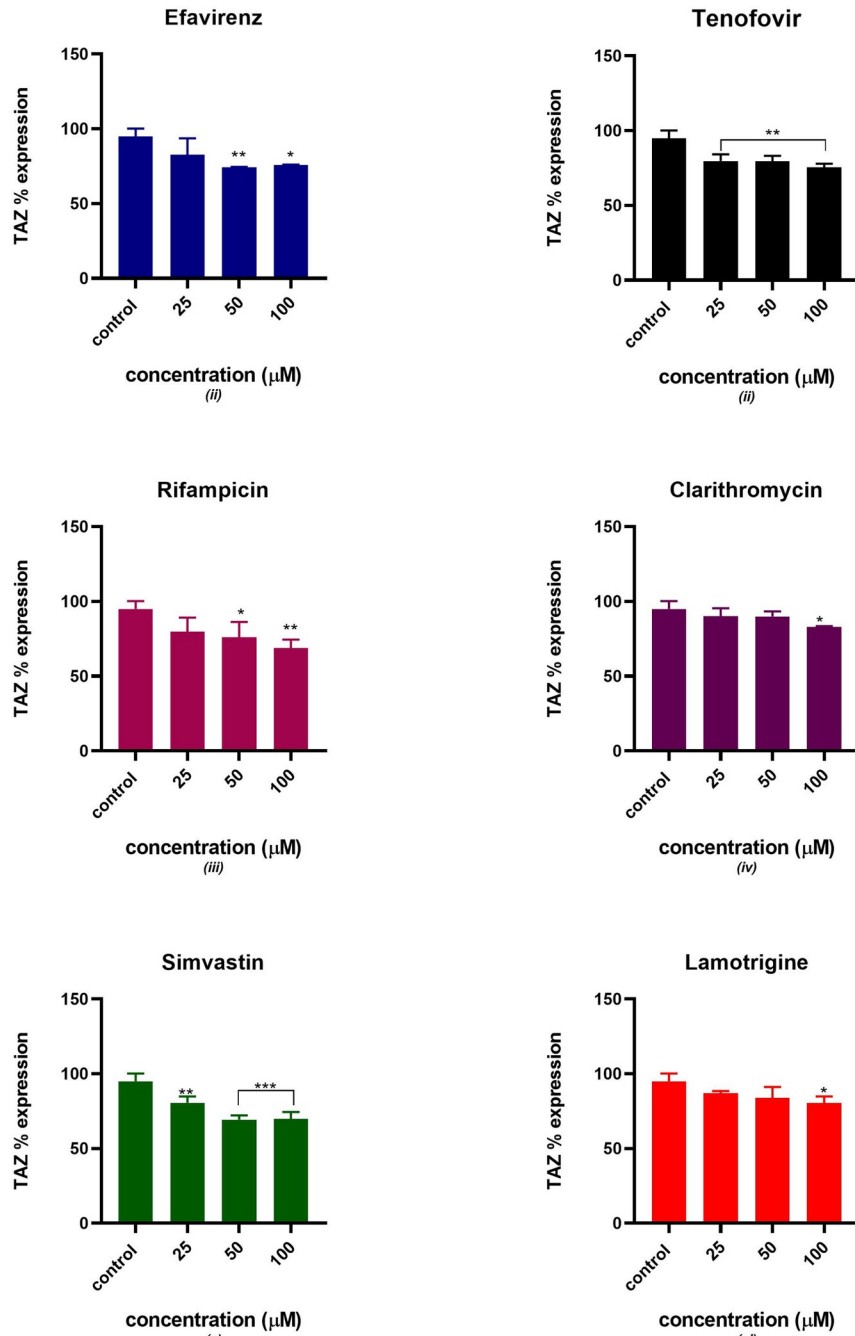

**Fig 6. Relative tafazzin protein % expression after 24 hours of exposure to potential mitotoxicants (25, 50, 100 μM).** The data are represented as mean ± SD represented with error bars, (n = 3), and the asterisk (*) represents the statistical difference between the test compounds and the control at (p < 0.05), (**) p<0.01, (***) p<0.001.

was no significant statistical difference between tenofovir and the control at p < 0.05. All the remaining drugs showed a significant statistical difference from the control at different concentrations except for (tenofovir and rifampicin) at 25 μM and simvastatin at 50 and 100 μM.

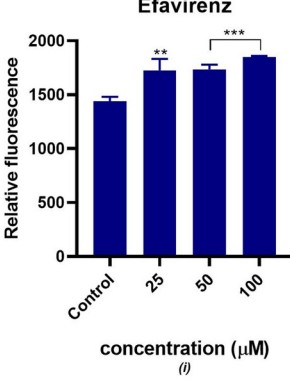

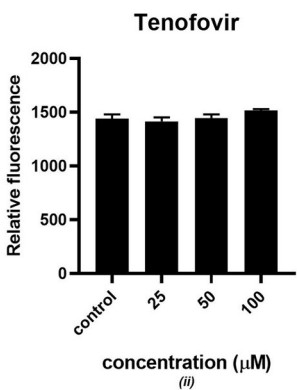

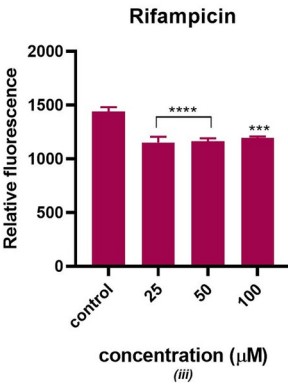

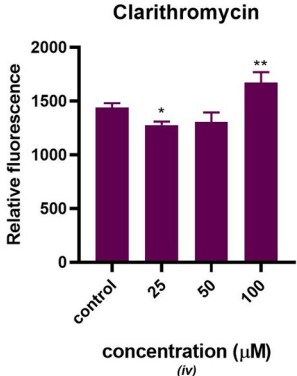

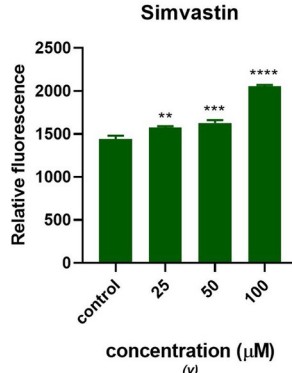

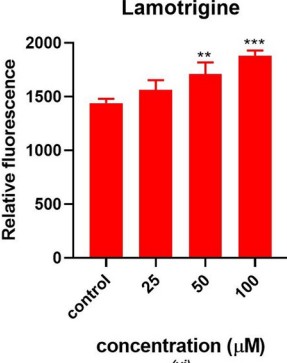

**Fig 7. Intracellular ROS production after 24 hours of exposure to potential mitotoxicants (25, 50, 100 μM).** The data are represented as mean ± SD represented with error bars, (n = 3), and the asterisk (*) represents the statistical difference between the test compounds and the control at (p < 0.05), (**) p < 0.01, (***) p < 0.001, (****) p < 0.0001.

## ATP bioluminescence

A commercially available bioluminescence kit was used to determine the ATP content. The luminescence signal was directly proportional to the concentration of ATP produced as displayed in Fig 10. rifampicin, simvastatin and clarithromycin significantly increased the concentration of ATP. Fig 10 shows a decrease in ATP production in the presence of lamotrigine,

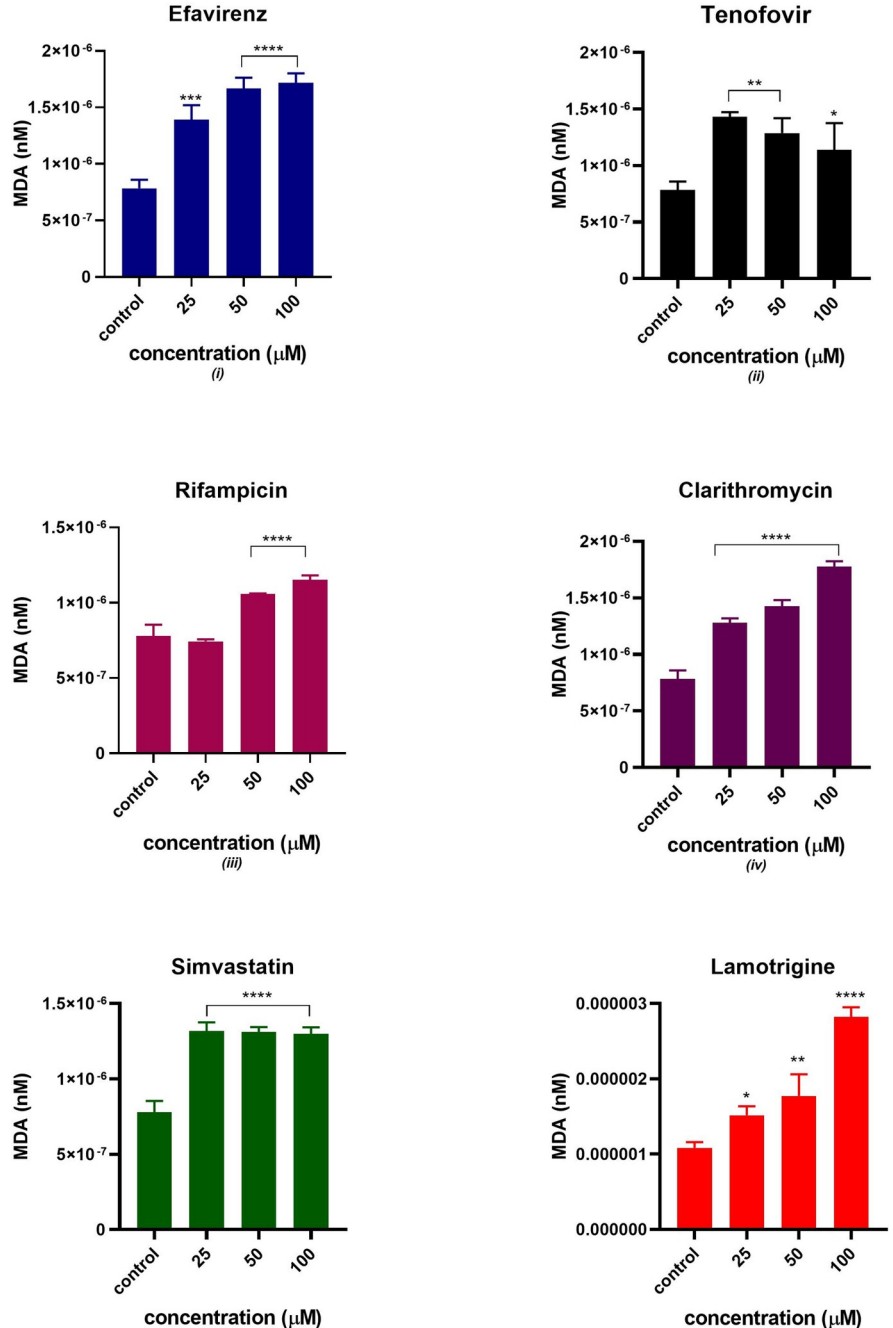

**Fig 8. Estimated medium MDA concentration after 24 hours of exposure to potential mitotoxicants at 25, 50 and 100 μM.** The data are represented as mean ± SD represented with error bars, (n = 3), and the asterisk (*) represents the statistical difference between the test compounds and the control at ($p < 0.0$), (**) $p < 0.01$, (***) $p < 0.001$, (****) $p < 0.0001$.

tenofovir. All six drugs showed a significant statistical difference from the control at different concentrations except simvastatin at 25 μM and clarithromycin at (25 and 50 μM).

**Efavirenz**

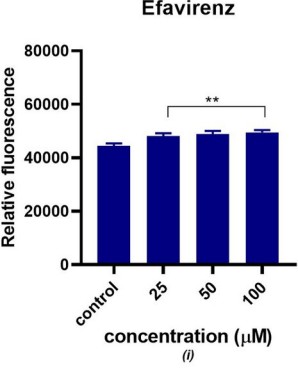

*(i)*

**Tenofovir**

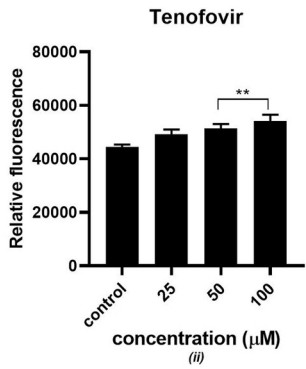

*(ii)*

**Rifampicin**

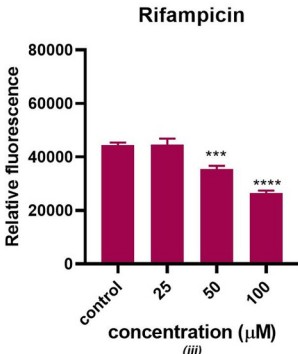

*(iii)*

**Clarithromycin**

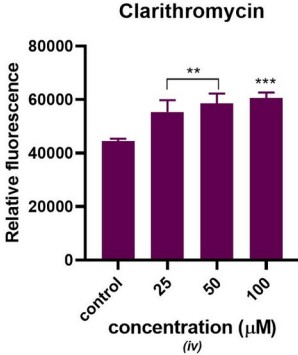

*(iv)*

**Simvastin**

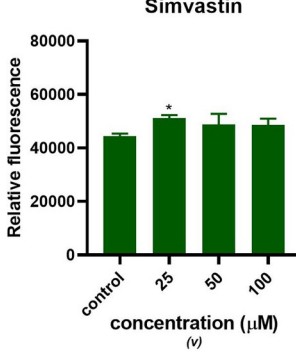

*(v)*

**Lamotrigine**

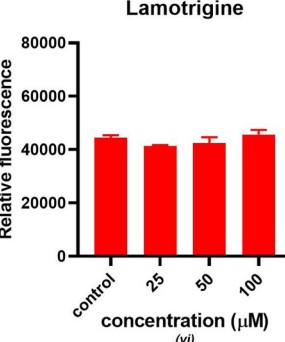

*(vi)*

**Fig 9. Mitochondrial membrane potential was evaluated after 24 hours of exposure to potential mitotoxicants (25, 50, 100 μM).** The data are represented as mean ± SD represented with error bars, (n = 3), and the asterisk (*) represents the statistical difference between the test compounds and the control at ($p < 0.05$), (**) $p < 0.01$, (***) $p < 0.001$, (****) $p < 0.0001$.

## Discussion

As alluded to, mitochondrial dysfunction has been reported to be one of the central causes of insulin resistance and its associated complications [10–12, 14]. Hence, it is imperative to perform studies to elucidate the link between drug-induced mitochondrial toxicity and the

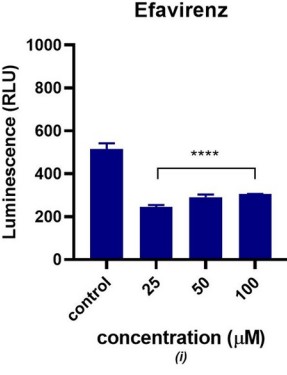

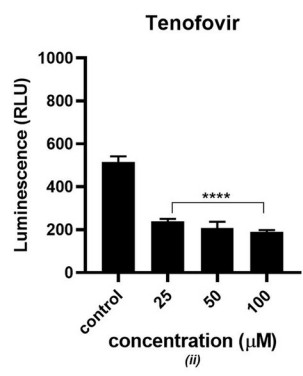

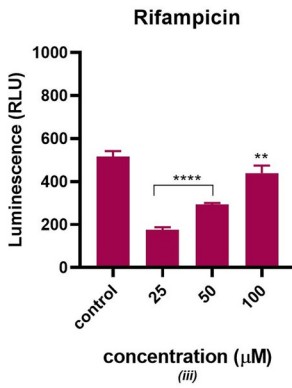

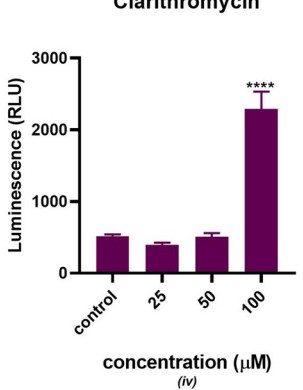

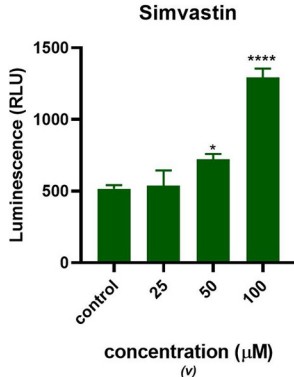

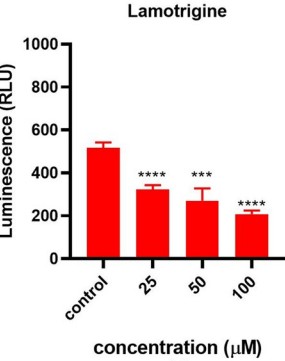

**Fig 10. Cellular ATP production after 24 hours of exposure to potential mitochondrial toxicants (25, 50, 100 μM).** The data are represented as mean ± SD represented with error bars, (n = 3), and the asterisk (*) represents the statistical difference between the test compounds and the control at $p < 0.05$, (**) $p < 0.01$, (***) $p < 0.001$, (****) $p < 0.0001$.

development of insulin resistance. In our study, we investigated six pharmacological agents that belong to different drug classes, particularly NNRTIs, NRTIs, antibiotics, statins, and anti-epileptics. C2C12 is an insulin-sensitive cell line that serves a beneficial role in insulin resistance mechanisms *in vitro* studies [16–18]. Skeletal muscle metabolism is mediated by insulin,

especially glucose uptake and synthesis of glycogen related to the PI3K/Akt signalling pathway [21]. Stimulation of the skeletal muscle by insulin activates a series of phosphorylation and P13K/Akt which culminates into GLUT 4 translocation, leading to glucose uptake [16, 22, 23].

Efavirenz is an extensively prescribed non-nucleoside reverse transcriptase inhibitor (NNRTI) however, it has been associated with the development of dysglycaemia and dyslipidaemia [24–26]. Efavirenz is administered as a single dose of 600mg in adults resulting in a therapeutic plasma concentration between 3.17 to 12.67 μM [27]. The aetiology of these metabolic effects remains unclear however, it has been reported that it is associated with mitochondrial toxicity [27, 28]. Blas-García and colleagues 2010 observed that clinical concentrations of efavirenz in hepatic cells acutely inhibited mitochondrial function resulting in bioenergetic stress. The mechanism of action involved the accumulation of lipids in the cytoplasm mediated by AMPK activation [28]. However, the coadministration of efavirenz with abacavir and lamivudine alters some of its mitochondrial effects [28, 29]. Furthermore, a cohort study conducted by Karamchand et al. in South African adults observed that an ART regimen consisting of efavirenz is associated with a higher risk of developing new-onset diabetes compared to a regimen containing nevirapine. Our observations suggest that exposure of C2C12 to efavirenz halts insulin sensitivity as demonstrated by poor glucose uptake and reduced expression of insulin signalling pathway proteins. This impairment of insulin signalling corroborated with increases in oxidative stress status as evidenced by the increase in intracellular ROS and MDA. Our study did not observe a significant change in ATP production and mitochondrial membrane potential. This implies that efavirenz possibly induces mitochondrial impairment through excessive generation of ROS, decreased tafazzin expression and the accumulation of lipid peroxides. To address whether there is a correlation between the generation of mitochondrial ROS and the development of insulin resistance *in vivo*, Anderson and colleagues administered a potent antioxidant peptide, SS31 which specifically targets the mitochondria [30]. The evidence suggested that the coadministration of SS31 in rats that were on a 6-week high-fat diet reduced the emission of mitochondrial $H_2O_2$ by 50%. The emitting potential of mitochondrial ROS was also stabilized, and the development of insulin resistance was mitigated. Efavirenz has also been reported to inhibit the secretion of adiponectin, a hormone involved in insulin sensitization and fatty acid breakdown. Taken together, these findings underscore the significance of mitochondrial dysfunction induced by efavirenz in the development of insulin resistance in insulin-sensitive tissues.

Various clinical trials have suggested that tenofovir disoproxil fumarate (TDF) used as pre-exposure prophylaxis (PrEP) to prevent HIV infection either in combination with emtricitabine (FTC) or not, has a good safety profile [31, 32]. TDF has shown favourable metabolic profiles, however, evaluating the specific adverse events induced by TDF/FTC on isolated metabolic parameters is challenging [33]. This is because various factors affect lipids and body composition including the HIV infection and the host response to the infection. Additionally, drugs that are part of the ARV regimen are likely to alter the pharmacokinetic properties of tenofovir. Previous studies suggested that mitochondria are subcellular target organelles of tenofovir toxicity [34–36]. In this study, the glucose utilization slightly decreased as the concentration of tenofovir increased, this can be linked to the decrease in GLUT 4 translocation and expression observed in Figs 3 and 4. These observations suggest that GLUT 4 expression and translocation can potentially be inhibited by tenofovir resulting in decreased insulin-stimulated glucose uptake. Additionally, the expression of Akt gradually decreased as the concentration of tenofovir increased as indicated in Fig 5. Thus, the downregulation in insulin-stimulated glucose uptake can be due to disruption of the chain of events in the insulin signalling pathway. Interestingly, tenofovir slightly decreased the expression of tafazzin, a non-specific phospholipid–lysophospholipid transacylase protein which is responsible for cardiolipin

synthesis and remodelling [37]. Various *in vivo* studies reported that tafazzin deficiency leads to reduced cardiolipin concentration and partially dissociates protein super complexes of the inner mitochondrial membrane [38–41]. Compelling evidence suggests that tafazzin-deficient mitochondria exhibit functional modifications such as increased oxidative stress, partial uncoupling, and reduced membrane potential [38, 42, 43]. These observations insinuate that tenofovir induced oxidative stress and possibly depleted the expression of tafazzin thus, halting oxidative phosphorylation as evidenced by the diminished ATP production.

Rifampicin is a macrocyclic antibiotic which widely prescribed as a component of a multi-drug regimen for tuberculosis [44]. In this study, we observed that rifampicin exhibited a dose-dependent reduction in insulin-stimulated glucose uptake. Garg et al conducted an *in vivo* study using mice models and observed that rifampicin improved glucose homeostasis and reduced diabetic complications by inhibiting the formation of advanced glycation end-products in mice [45]. However, in our study rifampicin gradually decreased glucose uptake associated with reduction of GLUT 4 expression and translocation. Studies have reported that rifampicin exhibited antioxidant properties by elevating p-AMPKα levels during a state of oxidative stress [46]. Furthermore, Liang et al observed that rifampicin preserved mitochondrial function through the reduction of ROS production in rotenone induced models [47]. On the contrary, Li et al reported a reduction in mitochondrial membrane integrity, accumulation of ROS and a decrease in ATP synthesis in human hepatocytes exposed to rifampicin [48]. In our study, an acute increase in the generation of ROS was observed. However, the significant increase in MDA production and loss of mitochondrial membrane potential can be attributed to perturbed mitochondrial function. Additionally, oxidative stress has been associated with poor glucose uptake through suppression of Akt. Various studies have closely linked excessive generation of ROS and mitochondrial dysfunction in the presence of rifampicin [49–51]. Considering that tafazzin expression acutely decreased, this could be an indication of mitochondrial dysfunction. Accordingly, it is plausible that the modification in mitochondrial oxidative metabolism contributes to the observed disruptions in insulin sensitivity.

The macrolide clarithromycin is used to treat an array of bacterial infections [52, 53]. Our study demonstrated that exposure of clarithromycin to C2C12 cells induced oxidative stress which could partly explain mitochondrial hyperpolarisation observed. ROS can lead to mitochondrial injury through mitochondrial DNA injury and damage to mitochondrial proteins including OXPHOS and transporter proteins [12, 19, 49]. Oxidative stress is a highly sensitive indicator of alterations in mitochondrial membrane integrity. An increase in the generation of ROS has been implicated in the onset of mitochondrial impairment and some studies have reported ROS as the primary cause of diabetic complications [54]. Hyperglycaemia is associated with excessive generation of ROS [4, 8]. These observations suggest that clarithromycin potentially induces insulin resistance through oxidative stress and altering mitochondrial membrane integrity. Furthermore, the ROS can also have a detrimental effect on insulin signalling proteins thus leading to a decrease in Akt expression and GLUT 4 translocation and expression. Various studies reported that increased generation of mitochondrial ROS occurs via inhibition of mitochondrial ETC at complex III [55–57]. In agreement with these observations, tafazzin was significantly reduced which can allude to a decrease in the function of OXPHOS.

There have been reports of an existing relationship between the use of antipsychotics and the development of T2DM [15, 58]. However, the exact mechanism of action is not yet fully elucidated. Previous studies proposed mechanisms for antipsychotic-induced DM such as weight gain and decreased insulin secretion in pancreatic ß cells by promoting cellular apoptosis [58–60]. Our study observed a slight decrease in insulin-stimulated glucose uptake as the concentration of lamotrigine increased which could be attributed to other affected GLUT

transporters considering that GLUT4 translocation was increased. In addition, Akt expression increased as the concentration of lamotrigine increased. Figs 7 and 8, suggest that lamotrigine induces oxidative stress. Various studies established that MDA is generated during lipid peroxidation and is a biomarker of oxidative stress. Lipids and lipoproteins are the major peroxidation targets in biological membranes thus, lipid peroxidation assays are commonly used to estimate oxidative status *in vitro* [20, 61, 62]. Houstis and colleagues reported that the generation of ROS in cultured 3T3-L1 adipocytes preceded insulin resistance and ROS scavenging improved insulin sensitivity [63]. The evidence suggested that ROS is the primary cause of insulin resistance. These findings suggested that the potential mechanism for lamotrigine-induced insulin sensitivity dysregulation is elevated levels of oxidative stress.

Simvastatin, a cholesterol-lowering agent has been recognised to impair insulin sensitivity [64–67]. Indeed, our study demonstrated impaired insulin signalling. Previous studies reported that the mitochondria play a significant role in statin-induced myopathies due to mitochondrial ubiquinone (CoQ10) depletion [68–70]. Interestingly, we observed a dose-dependent increase in cellular ATP synthesis in the presence of simvastatin. However, there was a slight increase in the maintenance of mitochondrial membrane potential as the dose increased. Fig 9 displays the effectiveness of simvastatin in maintaining MMP. These observations insinuate that the increase in ATP production observed is due to the effectiveness of simvastatin in maintaining MMP. Mitochondrial membrane potential maintenance is essential for preserving cellular homeostasis and ATP production [71]. The protective effect exhibited by simvastatin may be dependent on the mitoK$_{ATP}$ channels. Steven P Johns et al suggested that the protective effect of simvastatin deteriorates when the mitoK$_{ATP}$ channel is blocked in cardiac myocytes [72]. An increase in the generation of intracellular ROS and MDA was observed as displayed in Figs 8 and 9 indicating an increase in oxidative stress. Cumulatively, these observations suggest that the presumed diabetogenic effect of simvastatin is dose-dependent. The use of an isolated skeletal muscle system proposes the potential effect of simvastatin to directly exert a protective effect in the absence of other cell types. Additionally, the evidence proposing a link between the use of simvastatin and the development of T2DM varies among populations. For instance, an increase in T2DM incidence was observed in geriatric patients and prediabetic individuals [73].

Collectively, this study may further consolidate links between drug-induced mitochondrial toxicity and impaired insulin sensitivity as illustrated in Fig 11. Using different drug classes could further suggest that mitochondrial toxicity emanates from a wide range of conventional agents. For these reasons, our stride to curb diabetes should consider acknowledging some conventional pharmacotherapies indicated for different ailments as risk factors for developing insulin resistance. The varying observations with previous studies can be ascribed to varying cell culturing conditions, particularly experimental parameters and/or remodelling approaches. However, this study drew its conclusion from *in vitro* models thus, some of the observations vary from clinical manifestations reported. Most *in vitro* models of human skeletal muscles lack a three-dimensional structure and mechanical guidance cues found *in vivo* in native tissues [74–77]. They rely on the conventional two-dimensional (2D) cell cultures, which leads to the detachment of myotubes after a few days of myoblast fusion. Therefore, there is limited cell maturation and cessation in long-term experimental studies. Advanced 3D human skeletal tissue models improve upon 2D models thus allowing long-term culture and the cells can undergo functional tests such as measuring oxygen consumption, calcium handling, and metabolite production. An appreciation of the challenges and complexities associated with such measurements is crucial to avoid common experimental and analytical errors. Nevertheless, our findings do not preclude other probable mechanisms that mitochondrial toxicants can induce insulin resistance. Therefore, there is a need for further studies which

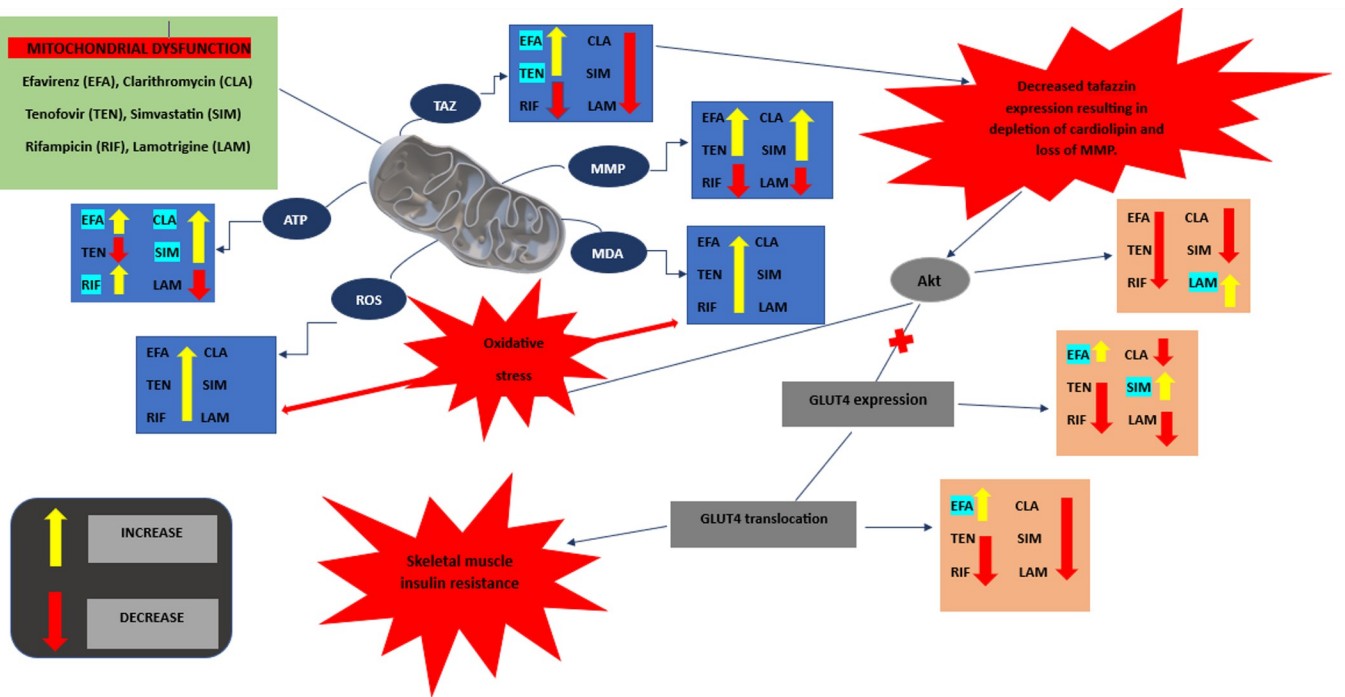

**Fig 11. An illustrative scheme of a proposed mechanism linking mitochondrial dysfunction induced via potential mitotoxicants and the development of insulin resistance.** Excessive production of ROS and MDA is observed in the presence of all the selected potential mitotoxicants resulting in an oxidative stress state. Tafazzin expression decreased suggestively indicating a depletion in cardiolipin content resulting in disrupted mitochondrial membrane permeability. Mitochondrial dysfunction results in diminished ATP production and disrupted mitochondrial membrane potential. The oxidative stress state and modified MMP disrupt oxidative phosphorylation in the insulin signalling pathway and affect the translocation and expression of proteins involved in the chain of events. Thus, mitochondrial dysfunction leads to increased ROS production and MDA, decreased TAZ expression and altered MMP resulting in insulin resistance in skeletal muscle.

seek to refine mechanisms associated with these mitotoxic agents in a context of disturbing glucose handling.

## Conclusion

In summary, our observations suggest that mitotoxicants can potentially induce insulin resistance emanating from mitochondrial dysfunction. However, further studies are required to fully define the clinical implications of mitochondrial dysfunction on the development of insulin resistance as demonstrated in our *in vitro* study.

## Supporting information

**S1 Data.**
(XLSX)

## Author Contributions

**Conceptualization:** Auxiliare Kuretu, Ntethelelo Sibiya.

**Data curation:** Auxiliare Kuretu, Ntethelelo Sibiya.

**Formal analysis:** Auxiliare Kuretu.

**Funding acquisition:** Ntethelelo Sibiya.

**Investigation:** Auxiliare Kuretu.

**Project administration:** Ntethelelo Sibiya.

**Supervision:** Mamosheledi Mothibe, Phikelelani Ngubane, Ntethelelo Sibiya.

**Validation:** Mamosheledi Mothibe, Phikelelani Ngubane, Ntethelelo Sibiya.

**Visualization:** Auxiliare Kuretu.

**Writing – original draft:** Auxiliare Kuretu, Mamosheledi Mothibe.

**Writing – review & editing:** Phikelelani Ngubane, Ntethelelo Sibiya.

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
