## [Decision Letter · Decision Letter 0]

16 Jul 2024

PONE-D-24-02153Elucidating the effect of drug-induced mitochondrial dysfunction on insulin signalling cascade and glucose handling in C2C12 cell line in vitroPLOS ONE

Dear Dr. Sibiya,

Thank you for submitting your manuscript to PLOS ONE. After careful consideration, we feel that it has merit but does not fully meet PLOS ONE’s publication criteria as it currently stands. Therefore, we invite you to submit a revised version of the manuscript that addresses the points raised during the review process.

First and foremost, I apologize for the delay in responding. Your manuscript was sent to over 20 potential referees, but it proved quite challenging to obtain responses from two reviewers. Their comments are appended below.

Both reviewers found your study to be of significant interest; however, they have several concerns that need to be addressed before we can proceed further. In particular, one reviewer suggests additional experiments regarding the effects of drugs on insulin-treated myotubes, which I concur with.

We look forward to receiving your revised manuscript.

Kind regards,

Makoto Kanzaki, Ph.D.

Academic Editor

PLOS ONE

2. We note that your Data Availability Statement is currently as follows: [All relevant data are within the manuscript and its Supporting Information files]

Additional Editor Comments:

First and foremost, I apologize for the delay in responding. Your manuscript was sent to over 20 potential referees, but it proved quite challenging to obtain responses from two reviewers. 

Reviewers' comments:

Reviewer's Responses to Questions

**Comments to the Author**

1. Is the manuscript technically sound, and do the data support the conclusions?

Reviewer #1: Partly

Reviewer #2: Yes

2. Has the statistical analysis been performed appropriately and rigorously? 

Reviewer #1: Yes

Reviewer #2: Yes

3. Have the authors made all data underlying the findings in their manuscript fully available?

Reviewer #1: Yes

Reviewer #2: Yes

4. Is the manuscript presented in an intelligible fashion and written in standard English?

Reviewer #1: Yes

Reviewer #2: No

5. Review Comments to the Author

Reviewer #1: GENERAL COMMENTS

The study tested the effects of several compounds to alter mitochondrial function and induce insulin resistance. There were some English/grammar issues that a thorough re-read can fix. Although the approaches were appropriate and the data appeared to be carefully obtained, the significant issue was that the drug studies were not completed on insulin treated myotubes.

SPECIFIC COMMENTS

Page 17, Results, line 6-7. The sentence “However, efavirenz, lamotrigine and rifampicin (100 μM) demonstrated a moderated non-significant decrease in cell viability” should be re-written. If you have conducted statistics to show a trend then indicate the P value of the trend. A non significant difference means that there is no difference between the groups.

Page 17, Glucose (25 ,50 ,100 μM) – reposition the commas after the numbers to (25, 50,100 μM)

One missing set of experiments is that the drug incubations should also have been completed on insulin treated myotubes. That would paint a more complete picture of the effects of these drug concentrations on myotube glucose handling as a whole.

Reviewer #2: Through evaluating the intracellular mechanisms leading to the development of insulin resistance in the presence of various conventional pharmacological agents reported as potential mitochondrial toxicants in skeletal muscle cell line, the authors deciphered the effect of drug-induced mitochondrial dysfunction on insulin signalling cascade and glucose handling in C2C12 cell line in vitro. They used differentiated C2C12 preparations to be exposed to various concentrations of efavirenz, tenofovir, rifampicin, simvastatin, lamotrigine, and clarithromycin, separately, and then observed and measured the effect of drug-induced mitochondrial dysfunction on insulin signalling cascade. And they finally argued that the mitochondrial toxicants could potentially induce insulin resistance emanating from mitochondrial dysfunction. However, as indicated by the authors, the potential mitochondrial toxicants, efavirenz, tenofovir, rifampicin, simvastatin, lamotrigine, and clarithromycin have their own specific effects on insulin signalling cascade. Why is the same observation and measurement performed for all the mitochondrial toxicants?

6. PLOS authors have the option to publish the peer review history of their article (what does this mean?). If published, this will include your full peer review and any attached files.

Reviewer #1: No

Reviewer #2: No

---

## [Author Response · Author response to Decision Letter 0]

10 Aug 2024

Reviewer #1: GENERAL COMMENTS

The study tested the effects of several compounds to alter mitochondrial function and induce insulin resistance. There were some English/grammar issues that a thorough re-read can fix. Although the approaches were appropriate and the data appeared to be carefully obtained, the significant issue was that the drug studies were not completed on insulin treated myotubes.

We thank the reviewer for the comments. 

We have substantively revised the manuscript for English and Grammatical issues. 

. Our goal aimed to understand how exposure of cells to mitotoxic agents could lessen insulin’ effects. Hence our approach aimed to pre-treat the cells with these mitotoxic drugs, followed by insulin treatment. In this way, we could be able to observe changes in insulin sensitivity post-exposure to mitotoxic agents such as glucose uptake and GLUT4 translocation. We believe through this approach, we have been able in part to prove that drug-induced mitochondrial dysfunction (using 6 separate classes of drugs) does affect insulin sensitivity as evidenced by a decrease in insulin-stimulated glucose uptake, AKT, and GLUT 4 translocation. We believe the experiment proposed by a reviewer could be a significant stand-alone study, as it is a different approach, considering that the cells would be pre-treated with insulin following by mitotoxic agents. As such, preliminary experiments will also be vital to establish time-points where one can observe the effects. We welcome the reviewer’s comment, however from what we have outline above together with logistical issues associated studies of this nature, we believe this can be conducted as a follow-up study. We would like to believe the current observation broadly shed light on the effect of mitochondrial dysfunction on insulin effect.

SPECIFIC COMMENTS

Page 17, Results, line 6-7. The sentence “However, efavirenz, lamotrigine and rifampicin (100 μM) demonstrated a moderated non-significant decrease in cell viability” should be re-written. If you have conducted statistics to show a trend then indicate the P value of the trend. A non significant difference means that there is no difference between the groups.

We thank the reviewer for this comment, this has been attended to as suggested

Page 17, Glucose (25 ,50 ,100 μM) – reposition the commas after the numbers to (25, 50,100 μM)

We thank the reviewer for this comment this has been fixed

One missing set of experiments is that the drug incubations should also have been completed on insulin treated myotubes. That would paint a more complete picture of the effects of these drug concentrations on myotube glucose handling as a whole.

We thank the reviewer for this insightful comment. The goal for this study was to broadly demonstrate how mitochondrial dysfunction through drugs affects insulin signalling pathway and glucose handling. Hence we aimed to understand how pre-exposure of cells to mitotoxic agents could lessen insulin effects. Our approach therefore aimed to pre-treat the cells with these mitotoxic drugs, followed by insulin treatment, after which measurements of parameters were conducted. In this way, we could be able to observe changes in insulin sensitivity post-exposure to mitotoxic agents. We believe through this approach, we have been able in part to prove that drug-induced mitochondrial dysfunction (using 6 different classes of drugs) does affect insulin sensitivity as evidenced by a decrease in insulin-stimulated glucose uptake, AKT, and GLUT 4 translocation. We believe the experiment proposed by a reviewer could be a significant stand-alone study, as it is a different approach, considering that the cells would be pre-treated with insulin, following by mitotoxic agents. As such, preliminary experiments will also be vital to establish time-points where one can observe the effects. We welcome the reviewer’s comment, however from what we have outline above together with logistical issues associated studies of this nature, we believe this can be conducted as a follow-up study. 

Reviewer #2: Through evaluating the intracellular mechanisms leading to the development of insulin resistance in the presence of various conventional pharmacological agents reported as potential mitochondrial toxicants in skeletal muscle cell line, the authors deciphered the effect of drug-induced mitochondrial dysfunction on insulin signalling cascade and glucose handling in C2C12 cell line in vitro. They used differentiated C2C12 preparations to be exposed to various concentrations of efavirenz, tenofovir, rifampicin, simvastatin, lamotrigine, and clarithromycin, separately, and then observed and measured the effect of drug-induced mitochondrial dysfunction on insulin signalling cascade. And they finally argued that the mitochondrial toxicants could potentially induce insulin resistance emanating from mitochondrial dysfunction. However, as indicated by the authors, the potential mitochondrial toxicants, efavirenz, tenofovir, rifampicin, simvastatin, lamotrigine, and clarithromycin have their own specific effects on insulin signalling cascade. Why is the same observation and measurement performed for all the mitochondrial toxicants?

We thank the reviewer for this insightful and interesting comment. Indeed, these agents could affect glucose metabolism through other affected pathways. However, what they have in common is disturbance in mitochondrial dysfunction. Although similar observations were reported, however, the extent on the effect was not the same, as some were more robust than others in affecting glucose handling and insulin signalling pathway. The same measurement was reported for the sake of uniformity and drawing comparison. The goal for now was to broadly demonstrate how mitochondrial dysfunction through commonly prescribed drugs affects insulin signalling pathway and glucose handling, therefore pose a risk of insulin resistance onset. We however believe this is a crucial point to take forward as we aim to further refine possible mechanisms utilized by these agents to alter glucose handling.

---

## [Decision Letter · Decision Letter 1]

2 Sep 2024

Elucidating the effect of drug-induced mitochondrial dysfunction on insulin signaling and glucose handling in skeletal muscle cell line (C2C12) in vitro

PONE-D-24-02153R1

Dear Dr. Sibiya,

We’re pleased to inform you that your manuscript has been judged scientifically suitable for publication and will be formally accepted for publication once it meets all outstanding technical requirements.

Kind regards,

Makoto Kanzaki, Ph.D.

Academic Editor

PLOS ONE

Additional Editor Comments (optional):

Congratulations!

Reviewers' comments:

Reviewer's Responses to Questions

**Comments to the Author**

1. If the authors have adequately addressed your comments raised in a previous round of review and you feel that this manuscript is now acceptable for publication, you may indicate that here to bypass the “Comments to the Author” section, enter your conflict of interest statement in the “Confidential to Editor” section, and submit your "Accept" recommendation.

Reviewer #1: All comments have been addressed

Reviewer #2: All comments have been addressed

2. Is the manuscript technically sound, and do the data support the conclusions?

Reviewer #1: Yes

Reviewer #2: Yes

3. Has the statistical analysis been performed appropriately and rigorously? 

Reviewer #1: Yes

Reviewer #2: Yes

4. Have the authors made all data underlying the findings in their manuscript fully available?

Reviewer #1: Yes

Reviewer #2: Yes

5. Is the manuscript presented in an intelligible fashion and written in standard English?

Reviewer #1: Yes

Reviewer #2: Yes

6. Review Comments to the Author

Reviewer #1: The concerns have been addressed. The only thing remaining is improving the image quality of the figures.

Figures 1 and 2, 7,9 and 11 had a pink background color that should be removed so that the background is white like the other figures.

Figure 11 has a low image quality and I could not read the words on the figure.

Reviewer #2: The authors addressed my raised comments. They also performed some needed revisions. No more comments.

7. PLOS authors have the option to publish the peer review history of their article (what does this mean?). If published, this will include your full peer review and any attached files.

Reviewer #1: No

Reviewer #2: No

---

## [Editor Report · Acceptance letter]

4 Sep 2024

PONE-D-24-02153R1 

PLOS ONE

Dear Dr. Sibiya, 

I'm pleased to inform you that your manuscript has been deemed suitable for publication in PLOS ONE. Congratulations! Your manuscript is now being handed over to our production team.

Kind regards, 

on behalf of

Dr. Makoto Kanzaki 

Academic Editor

PLOS ONE